# Calcium Kidney Stones are Associated with Increased Risk of Carotid Atherosclerosis: The Link between Urinary Stone Risks, Carotid Intima-Media Thickness, and Oxidative Stress Markers

**DOI:** 10.3390/jcm9030729

**Published:** 2020-03-08

**Authors:** Ho Shiang Huang, Pao Chi Liao, Chan Jung Liu

**Affiliations:** 1Department of Urology, National Cheng Kung University Hospital, College of Medicine, National Cheng Kung University, Tainan 70403, Taiwan; hshuang54@gmail.com; 2Department of Urology, College of Medicine, National Cheng Kung University, Tainan 70403, Taiwan; 3Department of Environmental and Occupational Health, Medical College, National Cheng Kung University, Tainan 70403, Taiwan; liaopc@mail.ncku.edu.tw

**Keywords:** atherosclerosis, kidney stone, hyperlipidemia, carotid intima-media thickness, hypercholesteremia, oxidative stress

## Abstract

Previous studies have suggested that kidney stone formers are associated with a higher risk of cardiovascular events. To our knowledge, there have been no previous examinations of the relationship between carotid intima-media thickness (IMT) and urinary stone risk factors. This study was aimed toward an investigation of the association between dyslipidemia, IMT, and 24-hour urinalysis in patients with calcium oxalate (CaOx) or calcium phosphate (CaP) stones. We prospectively enrolled 114 patients with kidney stones and 33 controls between January 2016 and August 2016. All patients were divided into four groups, according to the stone compositions—CaOx ≥ 50% group, CaP group, struvite group, and uric acid stones group. Carotid IMT and the carotid score (CS) were evaluated using extracranial carotid artery doppler ultrasonography. The results of a multivariate analysis indicated that a higher serum total cholesterol (TC) and low-density lipoprotein (LDL) were all associated with lower urinary citrate and higher CS in both the CaOx ≥ 50% and CaP groups. Higher serum TC and LDL were also associated with increased serum 8-OHdG levels in both groups. The levels of carotid IMT and CS in the CaOx ≥ 50% and CaP groups were all significantly higher than in the controls. These findings suggest a strong link between dyslipidemia, carotid atherosclerosis, and calcium kidney stone disease.

## 1. Introduction

A complete picture of the mechanisms involved in kidney stone disease is still not available although several factors and steps involved in the crystallization and stone formation in kidneys have been elucidated [1]. However, urologists are often puzzled to find patients with recurrent stone disease despite having “normal” 24-hour urine, and patients whose urine have been modified with drug therapy and nonetheless continue to form additional stones [2]. In fact, kidney stone disease is thought to be a systemic disease [3] and is linked with a number of other chronic diseases, such as cardiovascular disease (CVD) [4], chronic kidney disease [5], and diabetes mellitus [6].

In addition to free and fixed particle theories and the Randall plaque hypothesis of stone pathogenesis, a vascular theory of Randall plaque formation has also been suggested [7]. Although the precise mechanisms underlying the association between kidney stone disease and CVD remain to be elucidated, one potential risk factor could be atherosclerosis, as supported by the CARDIA study [8]. Production of reactive oxygen species (ROS) and development of oxidative stress is a common feature of CVD and calcium oxalate (CaOx) stone disease [9]. However, the association between atherosclerosis and Ca-containing kidney stone disease requires further investigation.

It is generally believed that atherosclerotic changes in the carotid artery mirror general atherosclerosis [10]. Ultrasound measurements of the intima media thickness (IMT) in the carotid arteries have been used as an indicator of coronary atherosclerosis [11]. The severity of carotid atherosclerosis can also be evaluated by the plaque score and maximum percentage stenosis on sonograms [12].

To test the hypothesis that calcium-containing stone disease (CaOx or calcium phosphate, CaP) is associated with atherosclerosis, we conducted a prospective study to identify the association of serum total cholesterol (TC), high-density lipoprotein cholesterol (HDL), and low-density lipoprotein cholesterol (LDL), on 24-hour urine chemical components and stone composition (CaOx and CaP). Then, the relationships among Ca-containing stone disease, serum cholesterol levels, and the severity of atherosclerosis were evaluated, as compared to the controls. The impact of serum major markers of inflammation and ROS were also evaluated and compared to the controls in order to define their association.

## 2. Materials and Methods

### 2.1. Study Population

This prospective study was approved by the ethics committee at National Cheng Kung University Hospital (NCKUH) (B-ER-103-400) on 27th, March, 2015. Patients admitted to the urologic ward from January 2016 to August 2016 for surgical intervention (endoscopic or percutaneous surgery) whose stones had been analyzed with infrared spectroscopy were eligible for enrollment. Written informed consent was collected from every enrolled patient. Patients younger than 18 years and patients on statins or stone-related medications were excluded. We also enrolled 33 healthy volunteers, matched by gender, as controls. None of the controls had urolithiasis, as indicated by imaging studies, and none had a previous stone history. The confounders were adjusted using multiple logistic regression analysis. The criteria of controls was according to our previous published study and other studies [3,13].

### 2.2. Classification of Patients by Stone Type

Each patient was classified based on his/her stone composition as follows [14]: (1) CaOx ≥ 50% group—CaOx component ≥ 50% with or without any CaP; (2) CaP group—containing any brushite (BR, CaHPO4·2H2O) or Ca3(PO4)2 with or without any CaOx; (3) struvite (ST) group when containing any ST; (4) uric acid (UA) group when containing any UA. At least one 24-hour urinalysis and a single fasting lipid profile evaluation were performed three weeks after stone removal. Some of the blood samples were sent to the NCKUH central laboratory for a lipid profile analysis, and the others were centrifuged at 3000 rpm for 10 min, and the serum was frozen at −20 °C, until further analysis.

### 2.3. The Protocol of Chemical Analyses

The 24-hour urine specimen was sent to the NCKUH central laboratory for pH, calcium (Ca), phosphate (P), uric acid (UA), and creatinine (Cr) determination. Urine oxalate and citrate levels were assayed with commercial ELISA kits (BioVision, Miloitas, CA, USA). Urine anti-lithic proteins—osteopontin (OPN) and Tamm-Horsfall protein (THP) were assayed using commercial ELISA kits (Cloud-Clone Corp., Houston, TX, USA). Urinary supersaturation with respect to CaOx and CaP was assessed using the index proposed by Tiselius et al. [15]—AP(CaOx)index by (2.09 × Ca0.84 × oxalate)/(citrate0.22 × Mg0.12) and AP(CaP)index by (0.432 × Ca1.07 × (pH−4.5)6.8)/Citrate0.20. Serum levels of total cholesterol (TC) and high- and low-density lipoprotein cholesterol (HDL-C and LDL-C) were measured at the Central Laboratory at NCKUH. The serum oxidative stress marker—8-hydroxy-deoxyguanosine (8-OHdG), the marker for nitrosative stress (nitrotyrosine (NT)), acute inflammation markers (interleukin 6 (IL-6)), neutrophil gelatinase associated lipocalin (NGAL), and adhesion molecule—vascular cell adhesion molecule 1 (VCAM1) were assayed with commercial ELISA kits (all from Cloud-Clone Corp., Houston, TX, USA). All markers were assayed from the serum specimen and adjusted according to their serum creatinine levels.

### 2.4. Carotid Artery Ultrasound Evaluation

Ultrasonographic scans of the extracranial carotid arteries were performed by well-trained, protocol-adherent technicians in the department of Neurology, NCKUH, for which the procedures were the same, as described previously [16]. The severity of carotid atherosclerosis in each subject was evaluated using three parameters—intima-media thickness (IMT), maximum percentage stenosis, and carotid score (CS) for IMT and plaques. The IMT was measured as the distance between the leading edges of the lumen-intimal and media-adventitia interfaces on the far wall of the common carotid artery (CCA). The segment extending from 10 mm proximal to the carotid bulb was scanned through three different longitudinal measurements (anterior, lateral, and posterior) [16]. The maximum IMT was the greatest thickness of the wall, including plaque lesions [16]. The maximum percentage stenosis was computed by measuring the residual lumen diameter and the original diameter at the maximal stenosis site and dividing the difference by the original diameter [17]. The scoring system for the carotid score (CS) was—0: IMT < 0.8 mm with no plaque; 1: IMT ≥ 0.8 mm with no plaque; 2: plaque < 50%; 3: when plaque ≥ 50% and < 70%; 4: plaque ≥ 70%; and 5: total occlusion.

### 2.5. Statistical Analysis

SPSS for Windows (version 17.0, SPSS Inc., Chicago, IL, USA) was used for all statistical analyses. All data are shown as a proportion, or as mean ± SD. Serum TC, HDL, and LDL levels were analyzed as dichotomous variables (abnormal vs. normal), to compare with the 24-hour urine chemical data and carotid artery atherosclerosis parameters, in order to understand their relationships to stone risk in CaOx ≥ 50% and CaP stone patients.

We used either the Mann-Whitney U test or a one-way analysis of variance (ANOVA), as appropriate, to compare the continuous variables. The Fisher’s exact test was used to compare the categorical variables with the controls for the 24-hour urine chemical parameters, carotid sonographic parameters, and serum markers. A multivariate analysis with a logistic regression was then performed to confirm which variables still significantly correlated with the abnormal cholesterol levels. The Pearson product-moment correlation method was used to determine and compare the correlation coefficients (*r* values) for the carotid artery ultrasonographic scan parameters and the serum 8-OHdG and other serum markers. Differences were regarded as significant at *p* < 0.05.

## 3. Results

There were 114 patients with kidney stone disease (including UPJ stones) and 33 controls included in this study (Table 1). Only thirty patients had pure stone compositions, whereas the remainder were mixed-type kidney stones. Ninety-four patients in the kidney stone group were within the Ca-containing stones. The major stone compositions in this cohort were CaOx (76.3%) and CaP (79.8%) (Appendix A). Compared to controls, patients in the Ca-containing stone group had no significant between-group differences in gender, BMI, and age, but other stone type patients (ammonium magnesium, uric acid, sodium urate) had significantly older, higher BMI, and more females (as compared to the controls). With respect to 24-hour urine chemicals, Ca-containing kidney stone patients had increased urine oxalate levels and significantly decreased urine levels of citrate and OPN, as compared to the controls.

Only the CaOx ≥ 50% group and the CaP group had significantly higher IMT and carotid scores (CS), as compared to the controls (Figure 1). However, there were no significant differences in the maximum percentage stenosis for each group, as compared to the controls (*p* = 0.074 in CaOx ≥ 50% group, *p* = 0.072 in CaP group, *p* = 0.934 in struvite group, and *p* = 0.184 in uric acid group). Positive correlations between CS and IMT (*r* = 0.539, *p* < 0.001 in CaOx group, and *r* = 0.560, *p* < 0.001 in CaP group) and between the CS and maximum percentage stenosis (*r* = 0.905, *p* < 0.001 in CaOx group, and *r* = 0.913, *p* < 0.001 in CaP group) were found in this study. Therefore, we chose CaOx ≥ 50% group and the CaP group to examine the impact of serum cholesterol level on 24-h urine chemistry and the changes in serum biomarkers for atherogenesis, in the Ca-containing kidney stone disease group.

In the CaOx ≥ 50% group, patients with higher serum TC levels (≥200 mg/dL) had higher urinary oxalate (51.4 ± 62.9 vs. 27.7 ± 11.8 mg/day), urine total protein (237.9 ± 214.6 vs. 125.9 ± 158.4), CS (2.4 ± 1.1 vs 1.6 ± 0.8), and lower urinary citrate (230.5 ± 114.5 vs. 373.8 ± 58.9 mg/day) than the controls (Table 2). They also had a lower urine volume (2.0 ± 0.7 vs 2.5 ± 0.6 L/day) than patients with low serum TC levels in the same group. Patients with lower serum HDL levels (≤40 mg/dL) in the CaOx ≥ 50% group had higher abdominal circumferences (95.9 ± 8.2 vs 87.8 ± 10.8 cm), urine total protein (345.3 ± 287.0 vs. 125.9 ± 158.4), IMT (0.67 ± 0.12 vs. 0.57 ± 0.10 mm), CS (2.6 ± 1.2 vs. 1.6 ± 0.8), and lower urinary citrate level (282.8 ± 133.6 vs. 373.8 ± 58.9) and urinary OPN levels (16.4 ± 15.5 vs. 24.9 ± 15.3 μg/day) than the controls. Patients with higher serum LDL levels (≥130 mg/dL) in the CaOx ≥ 50% group had increased abdominal circumferences, BMI, urinary oxalate, urine total protein, IMT, and CS, and lower urinary citrate levels than the controls (Table 2).

Patients with higher serum TC levels in the CaP group (Table 3) had higher AP(CaP) indices (764.9 ± 135.1 vs 4.9 ± 5.2), abdominal circumferences (95.7 ± 8.1 vs. 87.8 ± 10.8 cm), urine oxalate levels (45.1 ± 54.0 vs. 27.7 ± 11.8 mg/day), urine total protein (269.4 ± 246.4 vs. 125.9 ± 158.4), IMT (0.67 ± 0.14 vs. 0.57 ± 0.10 mm), CS (2.3 ± 1.1 vs. 1.6 ± 0.8), and lower urinary citrate levels (225.8 ± 95.1 vs. 373.8 ± 58.9 mg/day) than the controls. Patients with lower serum HDL levels in the CaP group (Table 3) had lower urinary OPN levels (15.9 ± 15.0 vs. 24.9 ± 15.3 μg/day) than the controls, in addition to having the same changes in the AP(CaP) index, abdominal circumferences, oxalate, urine total protein, IMT, CS, and citrate, as was found in those with higher serum TC levels in the same group. Patients with higher serum LDL levels in the CaP group (Table 3) had higher BMIs (26.5 ± 4.3 vs. 24.9 ± 3.5 kg/m^2^), in addition to the same findings found in those with higher serum TC levels in this group.

After adjusting by age and sex (Table 4), in the multivariate analysis, only urinary citrate levels (OR = 0.970, *p* = 0.003 in CaOx ≥ 50% group; OR = 0.970, *p* = 0.001 in CaP group) and CS (OR = 15.291, *p* = 0.013 in CaOx ≥ 50% group; OR = 2.964, *p* = 0.032 in CaP group) still had a significant association with higher serum TC in both groups. In patients with lower serum HDL levels, only CS (OR = 3.885, *p* = 0.015) and urinary citrate level (OR = 0.987, *p* = 0.022) in the CaP group remained significantly associated with lower HDL levels, when examined using a multivariate logistic regression analysis, after adjusting by age and sex (Table 4). In patients with higher serum LDL levels, urine levels of urinary citrate CS remained significantly associated with higher serum HDL in both groups, and the urine oxalate level was only significantly associated with higher serum LDL levels in the CaP group (OR = 1.030, *p* = 0.020) (Table 4).

In the univariate analysis results, it was found that patients with higher serum levels of TC and LDL had significantly higher serum levels of 8-OHdG and VCAM1 in both stone groups, as compared to the controls (Figure 2A,C); patients with high serum LDL levels in both stone groups had higher serum NGAL levels, as compared to the controls (Figure 2C). Higher serum IL-6 levels were found only in the CaP patients with higher serum TC levels (Figure 2A). Patients in the CaP group with lower serum HDL levels had significantly higher serum VCAM1 levels, as compared to the controls. In the multivariate analysis results after adjusting by sex and age, only the serum 8-OHdG levels remained significantly associated with high serum TC in both stone groups (Table 5). Lower serum HDL remained significantly associated with serum 8-OHdG and VCAM1 in the CaP group. In the higher LDL stone subgroup, serum 8-OHdG and NGAL remained significantly associated with high serum LDL levels in the CaOx ≥ 50% group, whereas serum 8-OHdG and VCAM1 remained significantly associated with high serum LDL levels in the CaP group.

The correlation between serum markers and 24-hour chemical parameters was analyzed using a Pearson product-moment correlation method. There were positive correlations found between serum 8-OHdG and VCAM1 (Figure 2D,E) and between serum 8-OHdG and IL-6, and a negative correlation was found between serum NGAL and urinary oxalate level (Figure 2F).

## 4. Discussion

A significant association with atherosclerosis was found in patients with Ca-containing kidney stones and dyslipidemia, as compared to the controls. In the multivariate analysis, after adjusting by age and sex, only urinary citrate and carotid scores remained significantly associated with high serum TC and high serum LDL levels in both stone groups and with low serum HDL in the CaP group. Urine oxalate levels remained significantly associated with high serum LDL levels in the CaP group. Serum 8-OHdG, a marker of oxidative damage of DNA, was significantly associated with high serum TC and LDL levels in both stone groups, and was associated with low serum HDL in the CaP group. The changes in serum 8-OHdG were positively correlated with serum VCAM1 and IL-6 levels in the high LDL subgroup with Ca stone disease. Significantly elevated serum NGAL was only found in the high serum LDL subgroup of Ca kidney stone patients and remained significantly associated with high serum LDL in the CaOx ≥ 50% group in the multivariate analysis, after being adjusted by sex and age, and was negatively associated with urinary oxalate levels.

### 4.1. A Comparison with Animal Studies

Rats fed with a cholesterol- and fat-rich experimental diet exhibited dyslipidemia, hyperoxaluria, hypercalciuria, dysproteinuria, and CaP nephrocalcinosis [18]. In our clinical study, we found hyperproteinuria, hyperoxaluria, hypocitrauria, but no hypercalciuria, in the higher serum LDL subgroup of both stone patients. One animal study found that rats fed with high cholesterol diet also exhibited high magnesuria [19]. However, our current clinical study failed to show hypermagnesuria in stone patients when compared to the controls. Hyperproteinuria was found in all Ca stone patients, and this result was consistent with an animal study [18] and another clinical report [20], but there was no significant differences between the kidney stone patient and the controls in the 24-h Ccr. Some authors have hypothesized that the retention of LDL and its potential oxidative modifications by intrinsic cells (exhibiting higher atherogenic potential) within the glomerulus might initiate a cascade of cellular events that are involved in the development of glomerulosclerosis [21].

### 4.2. Carotid Atherosclerosis is Associated with Ca-Containing Kidney Stone Disease

In present study, we found that patients with Ca-containing kidney stones had significantly higher IMT and CS, as compared to the controls. In 2011, Reiner et al. first demonstrated that young-to-middle-aged patients with self-reported kidney stone history were associated with subclinical carotid atherosclerosis, using a 20-year longitudinal cohort database (CARDIA study) [8]. No more studies have been conducted since to investigate the relationship between kidney stone and carotid atherosclerosis. To compare with the results of CARDIA study, the current study provided information about specific stone compositions and 24-hour urine chemistries in the patients with Ca-containing stones, and evaluated their associations with carotid atherosclerosis. In addition, UA stone formers were included in the CARDIA study, which has been proven to be associated with metabolic syndrome and thus might be a source of bias in the CARDIA study.

### 4.3. The Link Connecting Atherosclerosis and Kidney Stone Formation

Both animal [22,23] and clinical studies [3] have demonstrated that increases in oxidative stress in the kidneys of stone-forming patients and animals is indicated by the urinary excretion of ROS, the production of lipid peroxidation, and renal tubular enzymuria. An in vitro cell culture study showed that exposure to crystals, CaOx, and CaP leads to the development of oxidative stress [24]. Oxidative stress also plays a significant role in atherogenesis [25]. An in vitro study showed that oxidized low-density lipoproteins (ox-LDL) and oxidized lipoprotein induce O2- formation and apoptosis in endothelial cells and in the cultured human mesangial cells [26]. Human monocyte/macrophage studies have demonstrated that small, phagocytosable basic CaP crystals are more potent in terms of inducing the release of tumor necrosis factor-α, compared with larger particles [27] and that they can destabilize atherosclerotic plaques by initiating inflammation, causing vascular smooth muscle cell death and plaque rupture [28]. This scenario of atherogenesis is similar to the vascular theory of Randall plaque formation, which suggests that the repair of injured papillary vasculature in an atherosclerotic-like fashion, results in calcification near vessel walls that eventually erodes a carbonate apatite calculus into the papilla, through the renal papillary interstitium. The current clinical results showed elevated serum oxidative stress markers in the higher serum TC and LDL CaP and CaOx subgroups, as well as an association with serum VCAM1 and abnormal serum lipoproteins in the CaP group. Interestingly, the serum VCAM1 levels increased significantly in all of the abnormal CaP stone group serum cholesterol subgroups (i.e., high TC, Low HDL, and high LDL) and remained significantly associated with low serum HDL levels. Cybulsky et al. [29] found that VCAM1, not ICAM1, plays a more dominant role in the initiation of atherosclerosis and is a better marker for early atherosclerosis. These findings suggest that vascular-induced plaque formation might play a role in kidney stone formation, as theorized by Randall.

Apart from oxidative stress, chronic kidney disease (CKD) mineral and bone disorder (CKD-MBD) is a systemic disorder that alters calcium homeostasis and affects cardiovascular morbidity and mortality [30]. CKD-MBD involves the dysregulation of fibroblast growth factor 23 (FGF23)-Vitamin D (Vit. D)—parathyroid hormone (PTH) axis, which not only leads to kidney stone formation but is also associated with increased carotid IMT [31,32]. In our data, there was no significant difference in the level of PTH between the Ca-containing stone group and the control group (43.7 ± 15.7 vs. 44.6 ± 11.6 pg/mL, *p* = 0.664). In addition, no significant difference in the level of urinary Ca between the two groups was found (Table 1). However, we lacked the data of FGF23 and Vitamin D. Further studies are warranted to determine the role of CKD-MBD in the link between atherosclerosis and kidney stone formation.

### 4.4. Circulating NGAL Might Derive from the Toxic Effect of Urinary Oxalate

NGAL gained considerable diagnostic and prognostic value in kidney disorders as a valuable marker of acute kidney injury [33]. In patients with coronary artery disease, levels of circulating NGAL reflect the degree of the inflammatory process [34]. In present study, increased serum NGAL levels were found only in the high LDL subgroup of CaOx ≥ 50% stone patients, which showed a negative correlation with the urine oxalate levels only (*r* = −0.285, *p* = 0.0349 in CaP and *r* = −0.333, *p* = 0.036 in CaOx ≥ 50% group), but had no correlation with the serum IL-6 levels. This result implies that patients with CaOx kidney stones are at constant exposure to inflammatory insults, which might be generated by the urinary oxalate level. Proximal tubule damage caused by high urinary oxalate levels in stone patients might cause NGAL to leak into the urine and further decrease the serum NGAL levels. Thus, we found a significant negative correlation between serum NGAL and urine oxalate levels in renal stone patients.

There are some limitations in the current study. First, this study was a prospective clinical trial but was still subject to potential selection biases. Second, our data did not provide any dietary information from both groups. Especially urinary citrate excretion, diets rich in animal proteins were associated with hypocitraturia and vegetarian rich diets associated with hypercitraturia [35]. Accordingly, our results, which demonstrated the association between high cholesterol and hypocitraturia, are likely to be related to dietary habits. Third, there is still controversy over the value of IMT and carotid in predicting cardiovascular and cerebrovascular risks. Some studies suggested the association between IMT and the cardiovascular risks is still inconclusive [36,37]. Finally, our study recruited a relatively small number of cases. In the future, we intend to collect more cases to prove our observations.

## 5. Conclusions

Calcium kidney stone formers are associated with increased carotid artery atherosclerosis. Patients with Ca-containing kidney stones have higher levels of serum LDL (69.7% in CaOx ≥ 50% group and 70.3% in CaP group). Through multivariate regression analysis it was found that the carotid score, serum 8-OHdG level, and urine citrate levels were significantly associated with Ca-containing stone diseases.

## Figures and Tables

**Figure 1 jcm-09-00729-f001:**
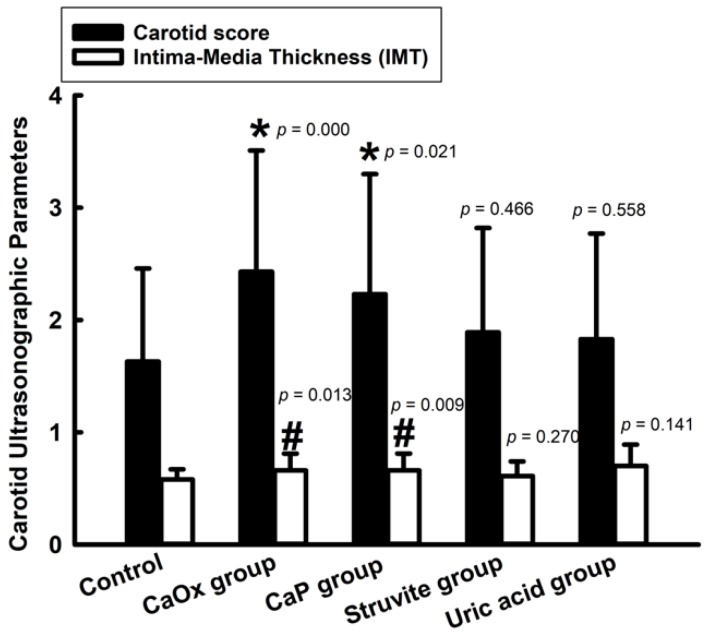
Carotid score and intima-media thickness (IMT) of carotid artery ultrasonographic scans in the different stone groups. *: *p* < 0.05 carotid score was significantly different when compared with the control. #: *p* < 0.05 IMT was significantly different when compared with the control. Abbreviation: CaOx, Calcium oxalate; CaP, Calcium phosphate.

**Figure 2 jcm-09-00729-f002:**
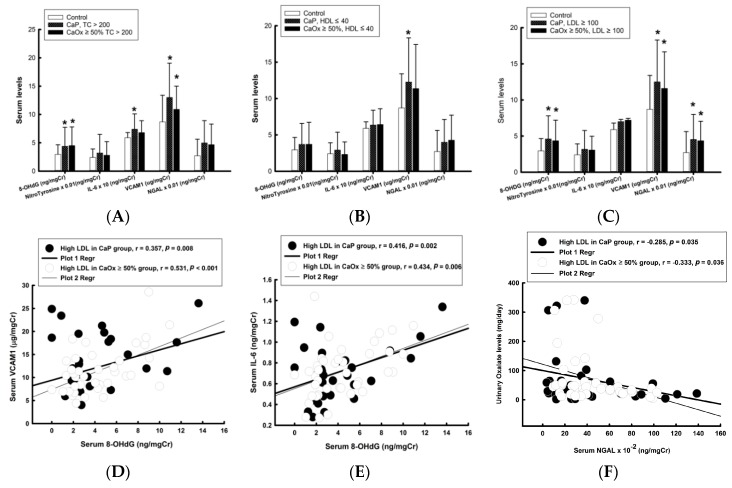
Serum markers of inflammation and oxidative stress compared to the controls in stone patients with CaOx and CaP, respectively. (The Nitro Tyrosine data was reduced 100X; IL-6 was magnified 10X, and neutrophil gelatinase associated lipocalin (NGAL) was reduced 100X from the original values in Figure 2A–C. (**A**), TC > 200, (**B**), HDL ≤ 40, (**C**) LDL ≥ 100. (**D**). Correlation between serum 8-OHDG and VCAM1; (**E**). Correlation between serum 8-OHdG and IL-6; (**F**). Correlation between urinary Oxalate level and serum NGAL level.

**Table 1 jcm-09-00729-t001:** Demographic data, 24-hour urinary chemicals, and stone composition in 147 patients. Data presented as mean ± SD.

Variables	Control (C)	Kidney Stone			
		Ca-Containing Stone		Other Stone Types ^#^	
	(*n* = 33)	(*n* = 94)	*p* Value (vs. C)	(*n* = 20)	*p* Value (vs. C)
Age	49.8± 12.9	52.4 ± 12.8	0.158	58.6 ± 13.0	0.021 *
BMI (Kg/m^2^)	24.9 ± 3.5	26.4 ± 4.4	0.065	26.9 ± 4.6	0.034 *
% Male	63.6	70.2	0.488	30.0	0.019 *
Comorbidity					
% Elevated fasting blood sugar	24.2	34.0	0.301	50.0	0.059
% Increased BP	24.2	32.9	0.353	50.0	0.235
% Large Abd. Circ.	9.1	23.4	0.077	25.0	0.124
% High TG level	18.2	35.1	0.071	30.0	0.330
% Reduced HDL-C	42.4	58.5	0.113	50.0	0.602
24-h urinalysis					
pH	6.0 ± 0.7	6.0 ± 0.7	0.542	6.2 ± 0.7	0.188
Volume (L)	2.1 ± 0.7	2.2 ± 0.7	0.316	2.3 ± 0.7	0.472
Calcium (mg/day)	191.6 ± 83.3	224.1 ± 144.7	0.509	147.4 ± 100.2	0.089
P (mg/day)	713.9 ± 225.5	745.9 ± 433.1	0.843	673.0 ± 246.9	0.373
Uric acid (mg/day)	666.7 ± 186.3	668.0 ± 289.2	0.873	550.7 ± 206.3	0.019 *
Mg (mg/day)	97.3 ± 31.4	106.7 ± 88.9	0.952	80.5 ± 38.7	0.015 *
Oxalate (mg/day)	27.7 ± 11.8	65.5 ± 83.2	0.006 *	52.16 ± 37.9	0.021 *
Citrate (mg/day)	373.8 ± 58.9	250.3 ± 124.6	<0.001 *	233.0 ± 131.9	<0.001 *
AP(CaOx)index	1.6 ± 2.1	2.4 ± 3.8	0.348	2.3 ± 3.7	0.882
AP(CaP)index	4.9 ± 5.2	30.1 ± 63.4	0.358	38.8 ± 66.8	0.301
OPN (μg/day)	24.9 ± 15.3	18.0 ± 13.9	0.024 *	12.4 ± 12.6	0.006 *
THP (μg/day)	18.6 ± 30.0	23.5 ± 36.9	0.473	36.7 ± 52.6	0.123

^#^: “Other stone types” includes ammonium magnesium (*n* = 11), uric acid (*n* = 6), and sodium urate (*n* = 3) in this study. **p* < 0.05 when compared with the control. Abbreviations: BMI, Body mass index; BP, Blood pressure; Abd. Circ., Abdominal circumference; TG, Triglyceride; HDL-C, High-density lipoprotein cholesterol; 24-h, 24-hour; P, Phosphate; Mg, Magnesium; AP(CaOx)index, ion-activity product of calcium oxalate; AP(CaP)index, ion-activity product of calcium phosphate; OPN, Osteopontin; THP, Tamm-Horsefall protein.

**Table 2 jcm-09-00729-t002:** Demographic, 24-hour urinary chemical data, and carotid score in the CaOx ≥ 50% group.

Variables	Control	CaOx ≥ 50% Group								
		TC (mg/dL)			HDL (mg/dL)			LDL (mg/dL)		
		<200	≥200	*p* Value	≦40	>40	*p* Value	<130	≥130	*p* Value
No. Pts	33	46	20		22	44		20	46	
Overall	†	159.3 ± 22.9 *	231.3 ± 28.7 *^,#^	<0.001	32.6 ± 4.9 *^,#^	56.2 ± 14.5	<0.001	78.4 ± 13.6 *^,#^	141.2 ± 34.2	<0.001
Age	49.8 ± 13.0	54.9 ± 9.1	49.1 ± 15.7	0.171	53.4 ± 11.4	53.0 ± 12.0	0.445	56.0 ± 7.8	51.9 ± 12.9	0.203
BMI (kg/m^2^)	24.9 ± 3.5	26.8 ± 4.5	26.5 ± 4.5	0.097	27.3 ± 4.3	26.4 ± 4.6	0.066	27.0 ± 3.8	26.6 ± 4.8	0.057
Abd. Circ. (cm)	87.8 ± 10.8	92.5 ± 11.6	92.3 ± 8.8	0.377	95.9 ± 8.2 *	92.9 ± 10.5 *	0.003	95.0 ± 9.0 *	93.5 ± 10.3 *	0.004
24-h urine chemistry										
pH	6.0 ± 0.8	5.9 ± 0.5	6.2 ± 0.8	0.532	6.0 ± 0.5	6.0 ± 0.7	0.95	6.0 ± 0.6	6.0 ± 0.6	0.901
Volume (L)	2.1 ± 0.7	2.5 ± 0.6 *^,#^	2.0 ± 0.7	**0.009**	2.5 ± 0.7	2.2 ± 0.7	0.16	2.4 ± 0.7	2.3 ± 0.7	0.365
Ca (mg)	191.6 ± 83.4	223.2 ± 129.9	222.0 ± 167.8	0.631	225 ± 145	221.8 ± 140.8	0.71	221.3 ± 91	223.6 ± 159.0	0.587
P (mg)	713.9 ± 225.5	777.7 ± 519.2	814.0 ± 316.6	0.46	824.9 ± 718	789.6 ± 292.3	0.148	873.7 ± 714.7	751.7 ± 323.9	0.926
Uric acid (mg)	666.7 ± 186.3	668.5 ± 251.3	675.7 ± 372.4	0.683	580.3 ± 301	715.8 ± 277.5	0.096	642.5 ± 279	682.9 ± 297.2	0.559
Mg (mg)	97.3 ± 31.4	101.2 ± 41.3	103.3 ± 70.3	0.648	99.5 ± 49.9	103.0 ± 52.4	0.943	103.2 ± 44.3	101.3 ± 54.4	0.823
Oxalate (mg)	27.7 ± 11.8	81.4 ± 92.4 *	51.4 ± 62.9	0.001	77.7 ± 115	70.9 ± 73.3 *	<0.001	73.9 ± 90.8 *	73.2 ± 90.8 *	0.003
Citrate (mg)	373.8 ± 58.9	272.5 ± 129.2 *	230.5 ± 114.5 *	<0.001	282.8 ± 133.6 *	245.0 ± 119.7 *	<0.001	223.5 ± 123.4 *	271.9 ± 124.7 *	<0.001
AP(CaOx) index	1.6 ± 2.1	2.5 ± 2.6	2.3 ± 3.3	0.216	1.9 ± 1.9	2.7 ± 3.2	0.134	2.2 ± 1.7	2.5 ± 3.2	0.248
AP(CaP) index	4.9 ± 5.2	16.3 ± 47.0	53.6 ± 98.8	0.163	14.3 ± 39.6	36.6 ± 81.6	0.706	29.9 ± 70.9	28.0 ± 70.3	0.659
OPN (μg)	24.9 ± 15.3	19.8 ± 14.7	20.2 ± 14.8	0.258	16.4 ± 15.5 *	22.0 ± 13.8	0.046	14.9 ± 10.2	21.8 ± 15.6	0.089
THP(μg)	18.6 ± 30.0	29.3 ± 42.9	12.9 ± 18.9	0.428	23.8 ± 37.3	27.3 ± 41.7	0.685	34.7 ± 52.7	22.7 ± 34.0	0.684
Carotid artery ultrasonographic parameters										
IMT (mm)	0.57 ± 0.10	0.64 ± 0.17	0.66 ± 0.18	0.071	0.67 ± 0.12 *	0.64 ± 0.16	0.046	0.67 ± 0.11 *	0.65 ± 0.16 *	0.016
Carotid score	1.6 ± 0.8	2.3 ± 1.2 *	2.4 ± 1.1 *	0.009	2.6 ± 1.2 *	2.3 ± 1.1 *	0.002	2.6 ± 1.1 *	2.3 ± 1.1 *	0.003
Stenosis %	13.3 ± 14.6	19.8 ± 17.2	19.3 ± 17.0	0.197	21.1 ± 16.6	19.0 ± 17.3	0.188	23.3 ± 18.1	18.6 ± 16.5	0.138
Renal function										
24-h Ccr (mL/min)	113.9 ± 33.5	105.6 ± 50.7	110.5 ± 73.8	0.212	103 ± 55.3	109.3 ± 60.7	0.211	106.2 ± 57.7	107.6 ± 59.6	0.213

Data presented as means ± standard deviation. † indicates TC = 187.7 ± 28.2 mg/dL, HDL = 54.4 ± 12.8 mg/dL, and LDL = 124.4 ± 27.8 mg/dL in controls. Abbreviations: CaOx, Calcium oxalate; TC, Total cholesterol; HDL, High-density lipoprotein; LDL, Low-density lipoprotein; BMI, Body mass index; Abd. Circ., Abdominal circumference; Ca, Calcium; P, Phosphate; Mg, Magnesium; AP(CaOx)index, ion-activity product of calcium oxalate; AP(CaP)index, ion-activity product of calcium phosphate; OPN, Osteopontin; THP, Tamm-Horsefall protein; IMT, Intima-media thickness of carotid artery; Ccr: creatinine clearance; Stenosis%, maximum percentage stenosis of carotid artery. *p* values: compared by ANOVA among the control, low cholesterol and high cholesterol groups. * *p* < 0.05 when compared with the control. ^#^: *p* < 0.05 when compared between lower cholesterol and higher cholesterol groups.

**Table 3 jcm-09-00729-t003:** Demographic, 24-hour urinary chemical data, and carotid score in the CaP group.

Variables	Control	CaP Group								
		TC (mg/dL)			HDL (mg/dL)			LDL (mg/dL)		
		<200	≥200	*p* Value	≦40	>40	*p* Value	<130	≥130	*p* Value
No. Pts	33	62	29		28	63		27	64	
Overall	†	159.2 ± 24.2 *	226.3 ± 25.5 *	<0.001	33.5 ± 5.3 *	56.3 ± 13.8	<0.001	80.2 ± 14.0 *	139.6 ± 30.2 *	<0.001
Age	49.8 ± 13.0	54.2 ± 13.2	51.4 ± 14.8	0.385	52.9 ± 14.3	53.0 ± 13.1	0.519	54.2 ± 14.1	52.4 ± 13.2	0.323
BMI (kg/m^2^)	24.9 ± 3.5	26.5 ± 4.7	26.3 ± 4.1	0.157	27.0 ± 4.1	26.2 ± 4.7	0.081	26.6 ± 5.0	26.5 ± 4.3	0.006
Abd. Circ. (cm)	87.8 ± 10.8	94.0 ± 10.6 *	95.7 ± 8.1 *	0.006	95.9 ± 8.2 *	92.9 ± 10.5 *	0.008	92.7 ± 12.2	92.4 ± 10.3 *^,#^	<0.001
24-h Urine chemistry										
pH	6.0 ± 0.8	6.1 ± 0.7	6.2 ± 0.8	0.364	6.2 ± 0.5	6.1 ± 0.8	0.420	6.2 ± 0.8	6.1 ± 0.7	0.626
Volume (L)	2.1 ± 0.7	2.3 ± 0.7	2.1 ± 0.8	0.134	2.4 ± 0.8	2.2 ± 0.6	0.386	2.2 ± 0.7	2.3 ± 0.7	0.501
Ca (mg)	191.6 ± 83.4	230.3 ± 129.5	227.9 ± 181.7	0.397	230.1 ± 173.1	229.3 ± 135.5	0.625	217.7 ± 112.4	234.5 ± 160.0	0.627
P (mg)	713.9 ± 225.5	762.7 ± 474.0	718.5 ± 374.3	0.871	811.2 ± 661.4	742.2 ± 311.2	0.383	781.9 ± 652.1	734.6 ± 323.0	0.693
Uric acid (mg)	666.7 ± 186.3	690.9 ± 268.6	636.2 ± 333.7	0.222	641.9 ± 331.2	687.6 ± 271.7	0.561	656.6 ± 294.3	680.7 ± 290.4	0.673
Mg (mg)	97.3 ± 31.4	109.6 ± 101.6	101.7 ± 62.6	0.852	122.0 ± 146.9	100.4 ± 48.9	0.969	122.4 ± 148.4	100.6 ± 50.2	0.972
Oxalate (mg)	27.7 ± 11.8	87.2 ± 100.8 *	45.1 ± 54.0	<0.001	85.6 ± 110.1 *	66.8 ± 79.2 *	0.008	84.4 ± 94.8 *	67.6 ± 87.3 *	0.003
Citrate (mg)	373.8 ± 58.9	267.6 ± 130.4 *	225.8 ± 95.1 *	<0.001	260.0 ± 136.7 *	249.9 ± 113.8 *	<0.001	222.6 ± 129.8 *	265.1 ± 115.2 *	<0.001
AP(CaOx) index	1.6 ± 2.1	3.1 ± 4.7	1.9 ± 2.8	0.123	2.5 ± 3.5	2.7 ± 4.4	0.357	3.3 ± 5.7	2.4 ± 3.4	0.251
AP(CaP) index	4.9 ± 5.2	409.2 ± 84.7 *	764.9 ± 135.1 *	<0.001	505.8 ± 114.1 *	604.9 ± 161.8 *	<0.001	762.6 ± 194.4 *	504.0 ± 120.8 *	<0.001
OPN (μg)	24.9 ± 15.3	18.5 ± 19.7	17.5 ± 14.6	0.077	15.9 ± 15.0*	19.1 ± 13.5	0.034	14.6 ± 10.6 *	19.2 ± 15.0	0.041
THP(μg)	18.6 ± 30.0	23.1 ± 36.7	23.5 ± 36.6	0.709	21.9 ± 35.3	23.8 ± 37.2	0.702	24.7 ± 41.7	22.2 ± 34.3	0.709
Carotid artery ultrasonographic parameters										
IMT (mm)	0.57 ± 0.10	0.65 ± 0.14 *	0.67 ± 0.14 *	0.026	0.67 ± 0.16 *	0.65 ± 0.15 *	0.012	0.67 ± 0.15 *	0.65 ± 0.15 *	0.014
Carotid score	1.6 ± 0.8	2.4 ± 1.1 *	2.3 ± 1.1 *	0.004	2.5 ± 1.1 *	2.3 ± 1.1 *	0.002	2.5 ± 1.0 *	2.3 ± 1.2 *	0.003
Stenosis %	13.3 ± 14.6	20.6 ± 16.3	17.2 ± 18.1	0.150	21.3 ± 17.0	18.8 ± 16.9	0.172	22.7 ± 14.5	18.2 ± 17.8	0.146
Renal function										
24-h Ccr (mL/min)	113.9 ± 33.5	110.6 ± 57.8	102.8 ± 66.5	0.130	111.0 ± 59.5	106.7 ± 61.5	0.168	113.9 ± 66.9	105.5 ± 58.1	0.193

Data presented as means ± standard deviation. † indicates TC = 187.7 ± 28.2 mg/dL, HDL = 54.4 ± 12.8 mg/dL, and LDL = 124.4 ± 27.8 mg/dL in controls. Abbreviations: CaP, Calcium phosphate; TC, Total cholesterol; HDL, High-density lipoprotein; LDL, Low-density lipoprotein; BMI, Body mass index; Abd. Circ., Abdominal circumference; Ca, Calcium; P, Phosphate; Mg, Magnesium; AP(CaOx)index, ion-activity product of calcium oxalate; AP(CaP) index, ion-activity product of calcium phosphate; OPN, Osteopontin; THP, Tamm-Horsefall protein; IMT, Intima-media thickness of carotid artery; Ccr: creatinine clearance; Stenosis%, maximum percentage stenosis of carotid artery. *p* Value: compared by ANOVA among the control, low cholesterol and high cholesterol groups. **p* <0.05 when compared with the control. #: *p* <0.05 when compared between lower cholesterol and higher cholesterol groups.

**Table 4 jcm-09-00729-t004:** Risk factors in each stone group compared to the controls, using a logistic regression with abnormal serum cholesterol levels (adjusted by age and sex).

Groups	CaOx ≥ 50%			CaP		
	OR	95%CI	*p* Value	OR	95%CI	*p* Value
TC ≥ 200 vs. Control						
Urine Citrate	0.970	0.951–0.990	0.003	0.970	0.953–0.987	0.001
Abd. Circ.	1.085	0.976–1.206	0.131	1.045	0.963–1.135	0.294
Urine Protein	0.999	0.994–1.004	0.758	1.001	0.997–1.005	0.617
Carotid score	15.291	1.778–131.492	0.013	2.964	1.100–7.983	0.032
HDL ≦ 40 vs. Control						
Urine Citrate	0.992	0.982–1.002	0.127	0.987	0.976–0.998	0.022
Urine OPN	0.995	0.936–1.058	0.874	1.023	0.950–1.102	0.543
Urine Protein	1.004	0.999–1.008	0.097	1.003	0.999–1.007	0.105
Carotid score	1.941	0.761–4.947	0.165	3.885	1.297–11.632	0.015
LDL ≥ 100 vs. Control						
Urine Citrate	0.857	0.759–0.969	0.014	0.914	0.871–0.958	0.000
Urine Oxalate	1.043	0.990–1.099	0.115	1.030	1.005–1.057	0.020
Urine Protein	1.001	0.997–1.005	0.570	1.004	0.999–1.008	0.113
Carotid score	4.786	1.327–17.258	0.017	3.582	1.435–8.943	0.006

Abbreviations: OR, Odds ratio; CI, Confidence interval; TC, Total cholesterol; CaOx, Calcium oxalate; CaP, Calcium phosphate; Abd. Circ., Abdominal circumference.

**Table 5 jcm-09-00729-t005:** Serum markers of inflammation with abnormal cholesterol levels compared to the controls, using a logistic regression (adjusted by age and sex).

Groups		CaOx ≥ 50%			CaP	
	OR	95%CI	*p* Value	OR	95%CI	*p* Value
TC ≥ 200 vs. Control						
8-OHdG	1.622	1.015–2.592	0.043	1.614	1.096–2.375	0.015
VCAM1	1.230	0.977–1.547	0.078	1.181	0.999–1.398	0.052
IL-6	47.49	0.147–52.456	0.190	4.896	0.088–270.84	0.438
HDL ≦ 40 vs. Control						
8-OHdG	1.275	0.863–1.886	0.223	1.623	1.021–2.581	0.041
VCAM1	1.144	0.956–1.369	0.142	1.273	1.037–1.562	0.021
IL-6	0.969	0.012–177.07	0.989	0.205	0.004–41.17	0.689
LDL ≥ 100 vs. Control						
8-OHdG	1.670	1.152–2.422	0.007	1.694	1.206–2.378	0.002
VCAM1	1.149	0.978–1.351	0.092	1.164	1.012–1.338	0.033
NGAL	1.003	1.000–1.006	0.047	1.002	1.000–1.004	0.097

Abbreviations: OR, Odds ratio; CI, Confidence interval; TC, Total cholesterol; CaOx, Calcium oxalate; CaP, Calcium phosphate; 8-OHdG, 8-Hydroxydeoxyguanosine; VCAM1, Vascular cell adhesion molecule 1; NGAL, Neutrophil gelatinase associated lipocalin.

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
