# Peer review of "Calcium Kidney Stones are Associated with Increased Risk of Carotid Atherosclerosis: The Link between Urinary Stone Risks, Carotid Intima-Media Thickness, and Oxidative Stress Markers"

_jcm, 2020, doi:10.3390/jcm9030729_

Round 1

Reviewer 1 Report

Review comments on Ho Shiang Huang al. “Calcium kidney stones are associated with increased risk of carotid atherosclerosis: the link between urinary stone risks, carotid intima-media thickness and oxidative stress markers”

The authors measured risk factors for kidney stones and cardiovascular events, carotid intima-media thickness (IMT) and a set of biomarkers in a cohort of 114 patients with kidney stones and 33 healthy controls. They found some associations between total cholesterol, LDL levels, urine citrate and IMT.

The topic of this study is of broad clinical relevance, but the findings are not particularly novel: previous studies showed significant associations between urolithiasis and dyslipidemia, and between kidney stones and coronary artery calcifications. Most importantly, the study presents major methodological limitations.

MAJOR POINTS

  • The authors do not provide any data on the selection criteria for the healthy control group. This is a critical point, since the most significant findings of the study were obtained by comparing kidney stone formers with healthy controls. It is not surprising that a group a younger people with a more favorable cardiovascular profile displayed lower levels of IMT. This finding is likely to be independent of any factor strictly related to kidney stones. Thus, the main message of this article might be the effect of a selection bias.
  • Along the same line, the paper does not include any dietary information. The association between urinary citrate and cholesterol levels can probably be explained by differences in diet, which need to be evaluated.

MINOR POINTS

  • Page 2: “we conducted a prospective study to evaluate the impact of TC, HDL and LDL on 24h-h urine chemical components and stone composition”. The study design only allows to identify associations and the word “impact” (which implies causality) should be avoided.
  • 1: I suggest changing the arrangement of the histograms to highlight how the groups were statistically compared.

Reviewer 2 Report

Dear Editor,

The aim of this study was to test the hypothesis that calcium-containing stone disease is associated with atherosclerosis.

It is very an intriguing question but I am afraid that authors did not find an answer.

The main problem is methodology,

The selection of a control group is essential, but there is nothing in “methodology section” about control group. In abstract – they are “healthy” . In table 1 they are “patients” and 25% had hypertension. There is nothing about inclusion and exclusion criteria.

Authors have studied impact of “calcium-containing stones disease“ on atherosclerosis
- 1. It is very difficult to do it when subjects from control group had lower rate of hypertension, smaller abdominal circumference and higher HDL-C.
2.  In table 1 data concerning patients with calcium stones should be presented separately (other patients with stone disease has no changes in US examination - figure 1)
3. Authors tried to study  the link between calcium stones and atherosclerosis. They presented so many results. Why PTH, FGF23, vitamin D levels were not studied? FGF23 , PTH and  Vitamin D are good candidates to be a link between calcium and athersoclerosis. There is nothing in discussion about PTH, FGF23, vitamin D.

Table 1 – explain “others” – there were only two patients in this group , what was a stone composition?

Figure 1. - This figure is essential for this study and very interesting.  I think that this figure should be discussed in details.

In Results and Discussion authors presented so many data concerning markers of inflammation, oxidative stress, lipids etc. It is not clearly written. It is difficult to noticed what is important

Table 3 – No Pts in column “HDL” is probably wrong , isn’t it ?

Conclusion is speculative.

Round 2

Reviewer 1 Report

The review process did not significantly improved the quality of the article.

Reviewer 2 Report

I have no comments